# A Deep Learning Technique to Improve Road Maintenance Systems Based on Climate Change

Haitham Elwahsh [1,*], Alaa Allakany [1], Maazen Alsabaan [2], Mohamed I. Ibrahem [3,4] and Engy El-Shafeiy [5]

1 Computer Science Department, Faculty of Computers and Information, Kafrelsheikh University, Kafrelsheikh 33516, Egypt; alaa.ellakani@kfs.edu.eg
2 Department of Computer Engineering, College of Computer and Information Sciences, King Saud University, Riyadh 11543, Saudi Arabia; malsabaan@ksu.edu.sa
3 School of Computer and Cyber Sciences, Augusta University, Augusta, GA 30912, USA
4 Department of Electrical Engineering, Faculty of Engineering at Shoubra, Benha University, Cairo 11672, Egypt
5 Department of Computer Science, Faculty of Computers and Artificial Intelligence, University of Sadat City, Sadat City 32897, Monufia, Egypt; engy.elshafeiy@fcai.usc.edu.eg
* Correspondence: haitham.elwahsh@gmail.com

**Abstract:** Road maintenance systems (RMS) are crucial for maintaining safe and efficient road networks. The impact of climate change on road maintenance systems is a concern as it makes them more susceptible to weather events and subsequent damages. To tackle this issue, we propose an RMSDC (Road Maintenance Systems Using Deep Learning and Climate Adaptation) technique to improve road maintenance systems based on Deep learning and Climate Adaptation. RMSDC aims to use the multivariate classification technique and divides the dataset into training and test datasets. The RMSDC combines Convolutional Long Short-Term Memory (ConvLSTM) techniques with road weather information and sensor data. However, in emerging nations, the effects of climate change are already apparent, which makes road networks particularly susceptible to extreme weather, floods, and landslides. Therefore, climate adaptation of road networks is essential, especially in developing nations with limited financial resources. To address this issue, we propose an intelligent and effective RMSDC that utilizes deep learning algorithms based on climate change predictions. The ConvLSTM block effectively captures the relationship between input features over time to calculate the root-mean deviation (RMSD). We evaluate RMSDC performance against frameworks for downscaling climate variables using two metrics: root-mean-square error (RMSE) and mean absolute difference. Through real evaluations, RMSDC consistently outperforms approaches with a reduced RMSE of 0.26. These quantitative results highlight how effective RMSDC is in addressing maintenance systems on road networks leading to proactive road maintenance strategies that enhance traffic safety, reduce costs, and improve environmental sustainability.

**Keywords:** deep learning; road maintenance systems; climate change; sensors data; RMSDC technique; multivariate classification; ConvLSTM

## 1. Introduction

The development of resilience in communities that are susceptible to the impacts of climate change is crucial for effectively adapting to the changing climate. Small developing countries are already feeling the effects of climate change, and with the projected increase in temperatures, the situation is anticipated to deteriorate further [1]. However, adaptation to climate change can be costly and many developing countries lack the necessary resources [2]. Transportation networks are particularly vulnerable to extreme weather events caused by climate change, putting island nations at risk [3]. Significant investments in road network adaptation will be necessary to ensure the safety of their populations and the continuity of local businesses [4]. Investigating whether adaptation expenses can be

minimized while maintaining safe driving conditions and an uninterrupted traffic flow is essential. Deep learning, an artificial intelligence technique that uses reinforcement learning to teach artificial neural networks to perform complex tasks, has achieved superhuman intelligence in various challenging applications [5]. This makes it a suitable tool for complex systems like weather and traffic. Even if human-induced greenhouse gas emissions stopped today, the signs of climate change caused by human activity are already evident and expected to worsen [6]. In February 2018, the concentration of carbon dioxide in the Earth's atmosphere reached over 407 parts per million, the highest recorded in the past 650,000 years. Furthermore, the average global temperature has risen by 1.8 degrees Celsius since 1880 [7]. The long-term effects of human-induced climate change are still uncertain. However, they are predicted to include sea-level rise, heatwaves, more frequent and severe storms, altered precipitation patterns, and increased floods and droughts in some areas [8]. As the world continues to warm, these consequences will likely intensify and place more stress on social and ecological systems [9].

## 1.1. Effects of Climate Change on Road Systems

Road transportation is crucial in modern society and various industries, ranging from traditional tuk-tuks to advanced electric vehicles. Although road networks are significant contributors to greenhouse gas emissions, they are indispensable for realizing the objective of the Paris Agreement, which aims to limit global warming to below 2 degrees Celsius by the year 2100 [10]. However, as temperatures continue to rise, road networks are becoming increasingly vulnerable to the consequences of climate change [11–13]. These impacts include rising sea levels, intensifying extreme weather events, altered precipitation patterns, and frequent floods and droughts. Without proper precautions, these consequences could damage or destroy road networks, limiting accessibility and compromising the safety of road users [14]. Effective adaptation to climate change can enhance the resilience of road networks, particularly in developing countries. Innovative building materials, early warning systems, and improved maintenance techniques can all contribute to building resilience [2]. Maintaining road networks is a critical component of creating resilient roads, with the potential to reduce government spending in low-income countries significantly. Transportation agencies may need to adjust system maintenance practices to account for the effects of climate change, such as changes in average air temperatures and winter precipitation patterns. Such modifications may necessitate more durable detour routes, rapid maintenance patrols to address more frequent potholes and buckling problems, and winter maintenance adjustments. This paper aims to investigate the application of deep learning to intelligent road maintenance systems as a potential solution to adapt to climate change, particularly in developing countries. The experiment involves training deep reinforcement agents to comprehend how traffic and adverse weather conditions contribute to road deterioration, maintenance costs, and traffic flow. The study also aims to investigate possible variations in the performance of deep learning algorithms under different scenarios. The paper proposes creating a simulation environment that replicates the general state of a road, taking into account the impacts of traffic, extreme weather events, maintenance activities, and costs. The simulation includes regular traffic flow, the opening and closing of roads for maintenance, and a budget for maintenance activities. Three deep reinforcement agents interact with the simulator, and their performance is evaluated based on their ability to manage the road's maintenance budget, prevent unacceptable levels of road deterioration, and maintain traffic flow on the more extensive road network. The study will also include comparisons with two non-intelligent agents: random and hard-coded.

## 1.2. Contributions

The work presented here offers the following key contributions:

- The proposed RMSDC architecture is innovative for multivariate time-series interpretability in road maintenance, particularly for multiple time-step forecasts.

- The spatial and temporal attention mechanisms are jointly trained in a unified design to learn the temporal and spatial contributions. The domain knowledge for the Road Maintenance dataset is utilized to explain the learned interpretations.
- RMSDC achieves state-of-the-art prediction accuracy while remaining interpretable. In most evaluations, RMSDC outperforms the baseline models, while in a few instances it matches the forecast accuracy of the baseline models.

## 2. Related Work

Road infrastructure is crucial for nations' economic and social well-being, providing numerous benefits. However, it is essential to note that road construction, maintenance, and usage have substantial environmental impacts [14]. In light of recent warnings from the UN's International Panel on Climate Change (IPCC) about climate change, there is a growing consensus on the need for sustainable development of infrastructure systems, particularly transportation networks [15]. The focus has shifted towards finding sustainable methods to maintain and repair eroding pavement networks while minimizing costs, greenhouse gas emissions, and the use of non-renewable resources [16].

Traditional assessments of pavement repair options have primarily considered economic and technical factors, overlooking the environmental implications [17]. Air pollution and climate change are two significant environmental challenges, with human-made greenhouse gas emissions contributing to global warming and posing severe consequences for the environment, society, and economy [18]. Since the transportation industry is a significant contributor to air pollution, special attention must be given to its impact [19]. Furthermore, the projected increase in natural disasters due to global warming necessitates a greater emphasis on Maintenance and Rehabilitation (M&R) procedures, which may increase pollutant emissions [18].

Creating a viable long-term maintenance framework is essential for policymakers to avoid significant capital waste while maintaining an acceptable level of service, especially considering financial constraints [20]. To achieve this, prediction models are required to develop effective maintenance strategies. Numerous budget allocation models have been proposed to address project-level and network-level concerns [21]. Recent studies have focused on pavement performance forecasting using local datasets or Long-Term Pavement Performance (LTPP) data [22,23]. The cost-effectiveness of methods is crucial for both data collection and maintenance frameworks. Collecting data should be economically feasible for small businesses, and cost-effectiveness is necessary to prevent budget waste in maintenance frameworks. Engineers have been investigating which maintenance methods yield the best results, often employing cost-effectiveness or cost–benefit analysis to assess their techniques [24,25]. Maintaining pavements over a long period allows for more cost-effective and environmentally friendly options to be explored, leading to extended service life and reduced capital costs. Combining treatments and developing reliable schedules can significantly contribute to these outcomes [26–28]. Additionally, the timing of maintenance treatments is crucial, and developing maintenance schedules based on long-term deterioration research has been the focus of extensive studies [29–32].

Machine learning has shown great potential in civil engineering by utilizing models created from data to mimic human intelligence. Many studies have tried integrating machine learning with civil engineering to tackle diverse engineering challenges [33–36]. Deep learning, a popular technique, has been widely used for damage recognition, crack detection, and prediction [37–40]. Reinforcement learning has emerged as a recent method to maximize cost-effectiveness in determining ideal treatment schedules [32,41]. Reinforcement learning (RL) can be applied to overcome decision-making challenges in long-term maintenance. Pavement authorities face difficulties stemming from the intricate nature of pavement deterioration, the presence of alternative treatments, and the utilization of diverse pavement performance indicators. Machine learning techniques, particularly reinforcement learning, offer a sequential decision-making approach for large-scale simulations and optimization of long-term maintenance planning [32,42–50] as shown in Table 1.

**Table 1.** Earlier research on the use of AI for infrastructure management.

| No. of Ref. | Research | Method | Main Category |
|:-----------:|:--------:|:------:|:-------------:|
| [44] | | CNN | Building |
| [45] | | CNN | Road |
| [46] | Identifying damage to infrastructure assets | CNN | Water |
| [37] | | KNN | Bridges |
| [47] | | CNN | Power |
| [48] | | ML | Bridges |
| [41] | Timing of Maintenance and Rehabilitation | RL | Road |
| [32] | | RL | Road |
| [43] | | RL | Road |
| [29] | | GA | Road |
| [22] | | KNN | Road |
| [37] | Performance Forecast | ANN | Dam |
| [49] | | ANN | Sewer |
| [50] | | RNN | Power |

There is a growing interest in adapting super-resolution architectures based on deep learning techniques for statistical downscaling, driven by the climate system's spatiotemporal characteristics and underlying non-linear behavior. Vandal et al. [51] proposed DeepSD, which treats intricate precipitation data as a single image. DeepSD incorporates the super-resolution architecture SRCNN [52], based on convolutional neural networks, to effectively capture spatial relationships. Other researchers have suggested ResLap [53], which utilizes a super-resolution network based on the Laplacian pyramid [54] to enhance the quality of derived climate change estimates. Additionally, deep learning, specifically reinforcement learning, is employed to predict the maintenance planning of road assets by integrating Life Cycle Assessment (LCA) and Life Cycle Cost Analysis (LCCA).

## 3. Theory and Methods

Deep learning techniques are rapidly being applied to road maintenance systems to increase the accuracy of repair recommendations while lowering costs. This section provides a high-level overview of CNNs, RNNs, and LSTMs for time series analysis as they apply to road maintenance systems.

### 3.1. Time-Series and Automated Statistical Downscaling (ADS)

ADS refers to a collection of data points arranged in chronological order, and it finds application in various fields that involve temporal measurements. It can be either univariate, relying on a single parameter, or multivariate, depending on multiple parameters.

In statistical downscaling, the Autoregressive-Scaling Model with covariate selection and prediction (ASD) is used for downscaling precipitation, and two critical steps are involved.

Researchers can obtain insights into the geographical distribution of road surface characteristics and how they vary over time by discussing the spatial and temporal aspects separately. Analyzing spatial characteristics can assist in identifying regional patterns and potential hotspots of specific State categories, whereas analyzing temporal elements can reveal seasonal trends and daily variations in road surface conditions. This understanding is critical for developing effective road maintenance methods and adjusting to changing weather conditions, ultimately improving the resilience and safety of the road network.

Firstly, the classification of rainy and non-wet days (precipitation exceeding 1 mm) is performed, followed by the prediction of the total amount of precipitation specifically for rainy days. Subsequently, the anticipated precipitation can be expressed as follows:

$$E(Y) = R \times E(Y|R) \; where \; R = \begin{cases} 0, & if \; P(Rainy) < 0.5 \\ 1, & Otherwise \end{cases} \tag{1}$$

Rainy and non-rainy days are represented as a binary variable in the framework, denoted as $R$. The proposed framework [55] employs five pairs of classification and regression approaches to test its effectiveness.

The discrepancy between the actual and expected results is then quantified as the loss. A loss function is employed to measure this difference. This study utilized two loss functions: Mean Square Error (MSE) and Categorical Cross-Entropy.

$$MSE = \frac{1}{n} \sum_{i=1}^{n} (y_i - \widehat{y_i})^2 \tag{2}$$

$$CategoricalCrossEntropy = - \sum_{i=1}^{n} y_i \times \log \widehat{y_i} \tag{3}$$

In this context, the gradient computation for each parameter involves using the loss, with $y$ representing the real value and $\hat{y}$ representing the projected value. The gradients provide insights into the potential adjustments needed for the parameters. This process, known as backpropagation, occurs during the tuning phase or learning process. By carefully adjusting the network's weights, the loss over the entire dataset is minimized, thereby enhancing the model's generalization ability.

### 3.2. Time Series Analysis Using Convolutional Neural Networks (CNNs)

CNNs are a sort of neural network widely used for image analysis; however, they can also be used to analyze time series. A CNN can be used in time series analysis for road maintenance to extract features from time series data by applying filters that capture patterns in the data. The filters' output is then sent through pooling layers to minimize the input's dimensionality and identify the most critical characteristics. CNNs have been demonstrated to be helpful in time series analysis for road maintenance [56], mainly when dealing with big datasets of road photographs.

Subsequently, the feature maps obtained from the convolution process undergo further processing using an activation function, commonly the widely used Rectified Linear Unit (ReLU).

$$ReLu(x) = max(0, x) \tag{4}$$

The pooling layer is employed to reduce the size of the feature maps generated by the preceding convolutional layer. Figure 1 illustrates the different stages of the convolution process.

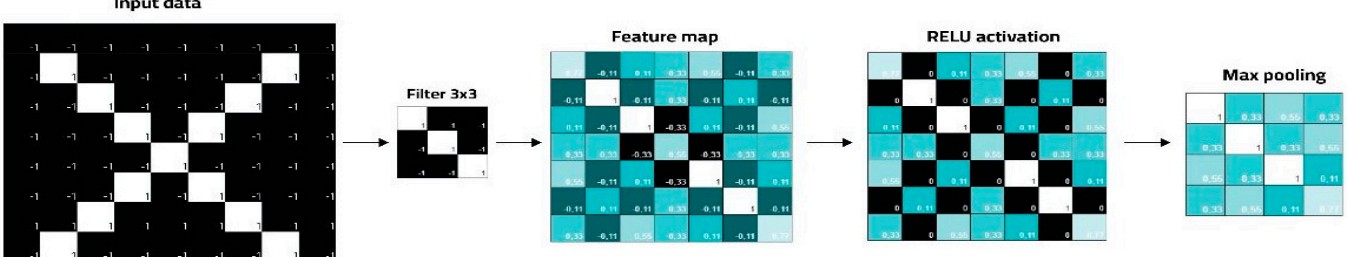

**Figure 1.** Steps in the convolution process.

The last layer utilizes the softmax activation function (as opposed to ReLU) to determine the probability of data belonging to a specific class.

The softmax function, when given an input vector denoted as z and containing K real numbers (corresponding to the output layer), is expressed as follows:

$$Softmax = (z)_i = \frac{e^{z_i}}{\sum_{j=1}^{k} e^{z_i}}, \ for \ i = 1, \ldots \ldots \ldots, k. \tag{5}$$

### 3.3. RNNs for Time Series

Analysis: RNNs are a sort of neural network that can handle sequential data, making them ideal for time series analysis of road maintenance. RNNs use feedback connections to maintain information over time, allowing them to capture temporal dependencies in data. Standard RNNs, on the other hand, are susceptible to the vanishing gradient problem, which limits their ability to detect long-term dependencies in data.

### 3.4. Time Series Analysis Using Long Short-Term Memory (LSTM) Networks

LSTMs are a sort of RNN that overcomes the vanishing gradient problem by controlling the flow of information through the network with a memory cell and numerous gating methods. The memory cell maintains information over time steps, while the gating mechanisms control the flow of information into and out of the cell. LSTMs have been demonstrated to be effective in time series analysis for road maintenance, mainly when dealing with data with long-term dependencies.

CNNs are used to analyze road images and identify different types of road surface damage in the proposed approach in "A Deep Learning Technique to Improve Road Maintenance Systems Based on Climate Change". In contrast, LSTMs are used to analyze climate data, estimate the probability of further damage, and prioritize maintenance actions. The suggested approach captures spatial and temporal connections in the data by combining CNNs and LSTMs, which can increase the accuracy of maintenance suggestions. Previous research has demonstrated the usefulness of deep learning approaches, such as CNNs and LSTMs, in time series analysis for road maintenance.

### 3.5. Convolutional LSTM Networks

LSTMs, a variant of RNNs introduced by Hochreiter et al. [57], differ from the traditional form by incorporating gates that enhance control over the gradient flow. These gates consist of the forget gate (F), which determines which components of the cell state can be discarded; the input gate (I), which regulates the addition or modification of components in the cell state; and the output gate (O), which determines the portion of the cell state to be output. The following equations and Figure 2 depict the three LSTM network modules:

$$f_t = \sigma\left(W_f \times [h_{t-1}, x_t] + b_f\right) \tag{6}$$

$$i_t = \sigma(W_i \times [h_{t-1}, x_t] + b_i) \tag{7}$$

$$o_t = \sigma(W_o \times [h_{t-1}, x_t] + b_o) \tag{8}$$

$$\hat{C}_t = tanh(W_c \times [h_{t-1}, x_t] + b_c) \tag{9}$$

$$C_t = f_t \times C_{t-1} + i_t \times \hat{C}_t) \tag{10}$$

$$h_t = o_t \times tanh(C_t) \tag{11}$$

These gates enable LSTMs to overcome the issue of vanishing gradients and maintain longer-term memory. In traditional RNNs, the gradient often diminishes significantly over long sequences, rendering the learning process ineffective.

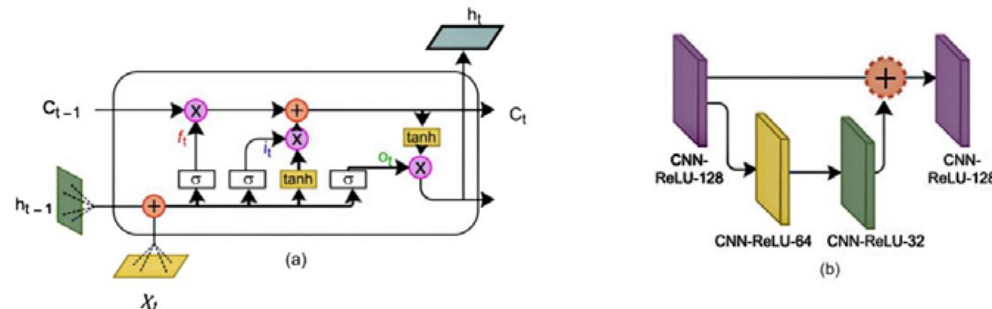

**Figure 2.** (**a**) Detailed ConvLSTM block. (**b**) Each block's variable adheres to the convention of listing the layer's name, its activation, and an integer to indicate how many kernels were utilized in that block.

The LSTM method for multivariate tuning comprises three stages: data conversion, LSTM modeling, and tuning. The data conversion module converts time-series data into supervised learning sequences and identifies the variable sets with the highest regression coefficients for the predictive value Y. The LSTM modeling module connects multiple LSTM perceptrons to construct an LSTM network. The tuning module iteratively adjusts the parameters based on root-mean-square deviations (RMSE) and updates the data for further training. The data conversion module tackles the challenge of multivariate problems with high dimensions by leveraging the periodicity of the data and reducing rows of data to a single row; it involves two processes: data preparation and data conversion. The second data conversion operation aims to combine multiple rows into one by transforming time-series data into supervised learning sequences based on the data's periodicity. Consequently, the operation defines the problem as finding a positive integer number. In the statistical downscaling problem, we consider the transfer function between high-resolution observations and coarse-resolution outputs using a spatiotemporal sequence of state variables as input. To address this, we propose an end-to-end trainable model called ConvLSTM SR (Convolutional LSTM Statistical Downscaling) by integrating fully connected LSTMs with convolutions. This model combines convolutional LSTMs with a super-resolution block.

The LSTM model used in our study, along with a simplified block diagram, is depicted in Figure 2. The LSTM model is constructed using the function: $C_t = f(X)$, where $h_t$ represents the variable value, $C$ denotes the climate state, $O_t$ represents the set of predictor variables on the $t_{\text{th}}$ day, and $C_{t-1}$ represents the variable value on the $(t-1)$th day.

## 4. RMSDC Technique Based on Multivariate Classification for Road Maintenance Systems and Climate Change

Road maintenance systems (RMS) are critical for guaranteeing the safety and efficiency of nations' road networks. Concerns have been made about the negative consequences of climate change on road maintenance systems, which make road networks more vulnerable to weather events and consequent damage. To solve this issue, we offer RMSDC (Road Maintenance Systems Using Deep Learning and Climate Adaptation), a new strategy that uses Convolutional Long Short-Term Memory (ConvLSTM) techniques with climate change forecasts to improve road maintenance systems.

We use a multivariate dataset in this study that includes road surface condition measurements (S1–S11), friction, temperature (Ta), moisture content (S7), road surface temperature (Tsurf), water content, vehicle speed, direction, geographical coordinates (latitude and longitude), elevation (height), and other relevant parameters. The categorical variable "State" describes the general condition of the road surface, with values ranging from 1 to 6 for Dry, Moist, Wet, Icy, Snowy, and Slushy.

We divide the dataset into training and test sets and use deep learning classification techniques to create RMSDC. The ConvLSTM architecture is used to capture temporal dependencies and interactions in data—notably, those that exist between input features

across time. The RMSDC model mixes road weather and sensor data, effectively adapting to climate change projections and calculating the root-mean deviation (RMSD).

The goal of the RMSDC (Road Maintenance and Smart Data Collection) is to create a dynamic framework that enables the development of rules for opening and closing roads based on changing weather and traffic data. These policies should ensure smooth traffic flow while keeping road maintenance costs within a predetermined budget. This section provides a detailed explanation of our proposed RMSDC technique and the suggested deep architecture.

This paper addresses the significant challenge statistical downscaling approaches face in effectively capturing spatiotemporal dependencies. We present a method that leverages recurrent convolutional LSTM to downscale ensembles of Smart Roads outputs, considering multiple starting conditions. Additionally, we propose a study approach to augment various state variables. To achieve this, we introduce additional variables related to weather conditions, such as surface temperature, wind speed, and air temperature. These weather variables play a crucial role, as heavy rain can lead to floods and landslides, which can impact the condition of the road.

The road is characterized by parameters such as estimated traffic volume, maintenance budget, and whether it is paved or unpaved. The present weather and traffic conditions influence the road's state. If the model keeps the road open, traffic can utilize it, whereas if it is closed, maintenance activities can be carried out to restore the road to its original condition.

Spatial Parameters—Latitude and Longitude: These geographic coordinates describe the measurement site's location. Examine how road surface conditions differ according to latitude and longitude, discovering places with distinct State categories.

Height: The elevation above sea level indicates the altitude of the place. Investigate how road surface conditions change with elevation, particularly for State categories such as Icy or Snowy, which higher elevations may alter.

Precision: The GPS measurement's precision reflects the location data's dependability. Consider whether the accuracy influences the classification of road surface conditions and the spatial distribution of State categories.

Temporal parameters—Date and Time: These parameters indicate the dataset's temporal aspect. Examine how road surface conditions change over time and on different days. Recognize patterns associated with seasons, weekdays, or specific times of day.

Seasonal Variations: Examine how the State categories fluctuate throughout the year. Examine whether certain road surface conditions, such as icy or snowy, are more common throughout specific months.

Examine how the road surface conditions change throughout the day. State types are more prevalent in the morning or at night.

Distance: The distance traveled by the vehicle, since the previous measurement indicates the sample frequency. Examine how the distance between data affects temporal resolution and road surface condition assessments.

We propose a Super Resolution approach based on Convolutional LSTM to statistically downscale climatic data from coarse-resolution to fine-resolution observation data. This approach considers the spatial and temporal dependencies between the target and auxiliary variables. In addition to station elevation data, we suggest incorporating physics-guided auxiliary variables that capture various state variables. Let $X_t$ represent the spatial data on the $t_{\text{th}}$ day, which includes climatic variables. Each $X_t$ represents a different climate variable, with an average of data points per unique coordinate over the observation period. Since the available data are limited for forecasting specific locations and times, we utilize them to interpret data from sensors and other sources that provide historical weather information.

$$X_t = [X_{t-T}, X_{t-(T-1)}, X_{t-(T-2)}, \ldots\ldots\ldots\ldots, X_{t-1}, X_t] \tag{12}$$

We introduce the problem that we aim to investigate and present the notations used in this paper. We are given $N$ time-series, denoted by $X = [X^1, X^2, \ldots\ldots, X^N]^T \in \mathbb{R}^{N \times T_x}$, representing the smallest form of the time-series data, as shown in Figure 3.

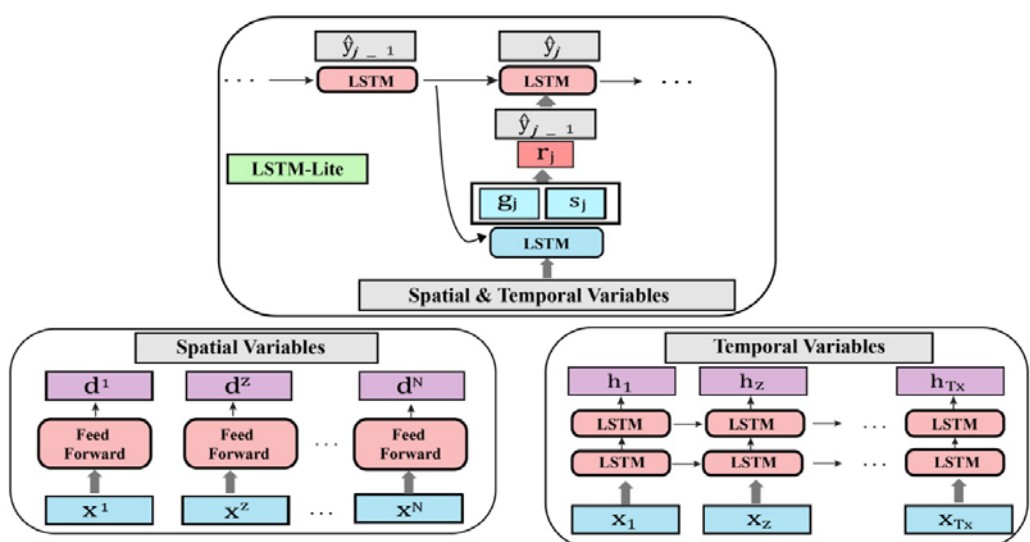

**Figure 3.** Using a time-series, the suggested model RMSDC is illustrated trying to compute the result.

Here, $T_x$ represents the length of the entire input sequence. Each $X^i = [x_1^i, x_2^i, \ldots\ldots\ldots\ldots, x_{T_z}^i]^T \in \mathbb{R}^{T_x}, i \in \{1, 2, \ldots\ldots, N\}$ corresponds to a time-series for each input variable $t \in \{1, 2, \ldots\ldots, T_x\}$, To represent the input variables for each time step $t \in \{1, 2, \ldots\ldots, T_x\}$, we use the notation $X_t = [x_t^1, x_t^2, \ldots\ldots\ldots\ldots, x_t^N]^T \in \mathbb{R}^N$. Thus, the compact form of all the time-series can alternatively be represented as $X = [X_1, X_2, \ldots\ldots, X_{T_T}]^T$. Similarly, we denote $y \in \mathbb{R}^{T_y}$ as a time-series where $y_j \in \mathbb{R}$ represents the output produced at time step $j$.

In the context of future time-series prediction, our goal is to develop a (non-)linear mapping, represented by a sequence model, to forecast $T_y$ future values of the output (univariate) time-series given the historical data for $T_x$ input (multivariate) time-steps. Mathematically, we define $F(.)$ as the mapping to be learned to obtain the forecasted solution $(\hat{y}_j)$ at output time-step $j$.

$$\hat{y}_j = F(\hat{y}_1, \hat{y}_2, \ldots\ldots\ldots, \hat{y}_{j-1}, X_1, X_2, \ldots\ldots\ldots\ldots, XT_x) \tag{13}$$

This study aims to develop unique mapping functions $F$ in Equation (13) that offer highly comparable or superior prediction accuracy while revealing the temporal and geographical relationships between input and output. Our approach enhances precise spatiotemporal interpretability, which is crucial in time-series prediction problems. This distinguishes our research from previous studies mentioned in the previous section. We utilize the spatial attention mechanism to assess the relative contributions of various input variables in multivariate time-series prediction. The inclusion of spatial attention in the encoding process has recently been proposed [58].

The spatial attention $\beta_t^i$ at time-step $t$ is computed as follows given the $i$-th attribute time-series $x^i$ of length $T_x$:

$$e_t^i = V_e^T tanh\left(W_e[h_{t-1}; C_{t-1}] + U_e X^i\right) \tag{14}$$

$$\beta_t^i = \frac{exp(e_t^i)}{\sum_{o=1}^{N} exp(e_t^o)} \tag{15}$$

The weighted input time-series at time $t$, $x_t$ is then substituted for the raw input time-series at time $t$, $\hat{x}_t$, and $\hat{x}_t$ is used as input to the encoder LSTM (function $f_1$) to compute the new states $h_t$ and $c_t$.

$$\hat{x}_t = \left[ \beta_t^1 x_t^1, \ \beta_t^2 x_t^2, \ldots\ldots\ldots\ldots, \beta_t^N x_t^N \right]^T \tag{16}$$

$$(h_t, c_t) = f_1(h_{t-1}, c_{t-1}, \hat{x}_t) \tag{17}$$

Following the encoder, it has been proposed to utilize the initial temporal attention method [59]. The attention weight for each hidden state of the encoder is determined by Equation (17) at the output time-step $j$ of the decoder.

$$\alpha_j^t = \frac{exp(\alpha_j^t)}{\sum_{l=1}^{T_x} exp(\alpha_j^l)}, \ s_j = \sum_{t=1}^{T_x} \alpha_j^t h_t \tag{18}$$

In this paper, we introduce the RMSDC technique based on a deep learning configuration. We comprehensively explain the RMSDC structure, including its mathematical formulations and concepts.

The detailed descriptions and processes of the RMSDC are presented in the subsequent sections, as depicted in Figure 4. The alignment between the output $y_j$ and input $x_t$ is determined by the probability $\alpha_j^t$. An alignment model, which is a feed-forward neural network function of $h_t$ and the previous decoder hidden state $h_t$, calculates the corresponding energy for $\alpha_j^t$. The temporal context vector $s_j$ serves as the input to the decoder at output time-step $j$. This method, commonly used in temporal interpretability studies, enables the computation of temporal attention weights [60].

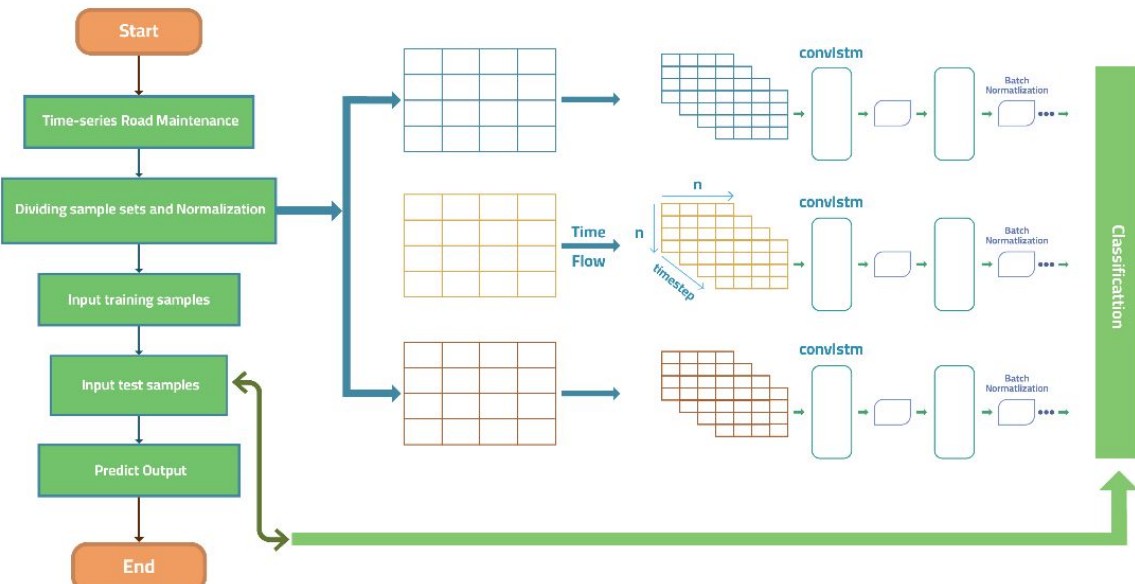

**Figure 4.** Flow chart for RMSDC based on multivariate.

*Dataset*

The dataset contains data for months, including weather conditions and measurements from numerous sensors relating to road conditions; environmental variables are also included in the data, such as in [61]. Each row of data represents a single measurement performed at a particular time and location for the Smart Road. The following are the data columns:

Day: The measurement was taken on this day.

Time (+01:00): The measurement time is adjusted for the local time zone (+01:00). S1–S3: Road surface condition measurements from three distinct sensors (S1, S2, and S3)

Friction: A measurement of the friction coefficient of the road surface, which indicates how slippery the road is, with 0.1–0.81 as the measured friction value.

Ta: The air temperature at the time and place of measurement.

S7: A sensor measurement of the road surface's moisture content.

Tsurf: The road's surface temperature at the time and location of measurement.

S9–S11: Road surface condition measurements from three distinct sensors (S9, S10, and S11)

Water: The amount of water on the road's surface at the time and location of meas urement.

Speed: The vehicle's speed at the time and location of the measurement.

The direction in which the vehicle was traveling at the time the measurement was taken.

The latitude of the site where the measurement was taken.

The longitude of the site where the measurement was made.

Height: The elevation above sea level where the measurement was taken.

Accuracy: The GPS measurement's precision.

Tdew: The temperature at the dew point at the time and location of measurement.

Friction 2: A second measure of the friction coefficient of the road surface that may be measured with a different method or sensor.

Distance: The distance the vehicle has traveled since the previous measurement.

Serial (RCM411): The serial number or identifier of the data collection device (data logger).

State: is a categorical variable that indicates the overall condition of the road surface—Dry, Moist, Wet, Icy, Snowy, and Slushy—with values from 1–6.

To examine more information from the dataset, we extract the days from Timestamp (changing this string to a DateTime object) and visualize the distribution of values each day. As illustrated in Figure 5, there were some variances in using values on different days.

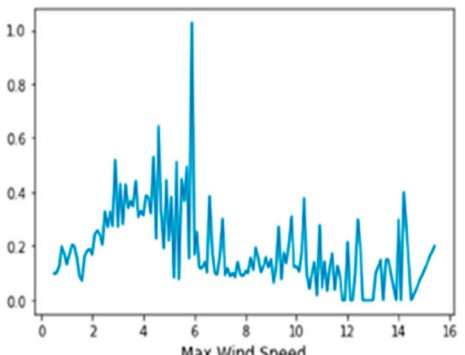 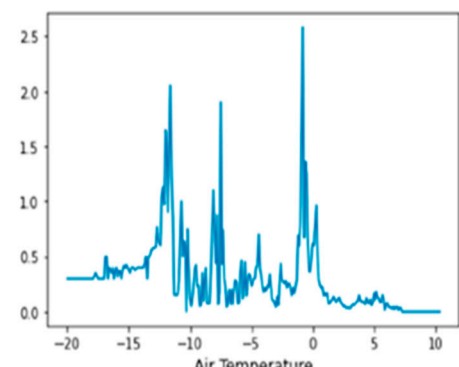

**Figure 5.** Distribution of the input values on different days.

We analyzed each data group by separating it using these spatial and temporal characteristics, allowing us to obtain insights on the spatial distribution of road surface conditions at different locations and the temporal patterns of road surface conditions across time. This divide will allow for more targeted and detailed investigations, which will help understand how road surface conditions vary across geographical regions and how they evolve over time.

RMSDC is an effort to apply the LSTM model to a time-series in both the spatial and temporal directions—based on geographical and sensor data, to be particular. We use an LSTM model throughout the spatial domain. Latitude and Longitude: Sort the data by latitude and longitude. These are data subsets prepared for distinct geographical regions. Height: The data are sorted by elevation (height above sea level) and temporal factors Date: Sort the data based on date values. Subsets of data are created for each unique date in the collection. Time: Sort the data based on time values. Data subsets are created for each time of day. Season: Divide dates into seasons (e.g., winter, spring, summer, and

autumn) and create data subsets for each. Distance: The distance between measurements is measured and recorded; the data are grouped based on the distance values. Subsets of data are created for various distance intervals, and the model is trained to learn the spatial and temporal structure. Figure 6 depicts an overview of RMSDC.

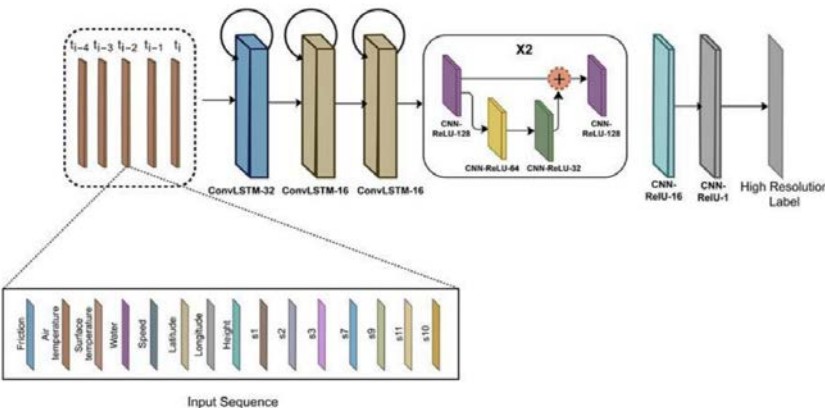

**Figure 6.** RMSDC based on convolutional LSTM structure.

## 5. Experiments and Discussion

The numerical solution of non-linear dynamical equations is of utmost importance in climate modeling, as it involves equations of state and the conservation of mass, energy, and momentum. These equations yield a range of state variables such as temperature, humidity, atmospheric pressure, wind velocities in three directions (zonal, meridional, and vertical velocities), and precipitation. However, employing higher temporal resolutions in running climate models or RMS increases computational expenses due to the non-linear relationship between spatial resolution and execution time. In this article, we leverage the coarse resolution outputs from the DIT4BEAR's Smart Roads Internship [61].

### 5.1. Evaluation Metrics

The evaluation metrics for the RMSDC model with ConvLSTM block for predicting road maintenance needs with climate change data depend on the specific objectives. Some suggested evaluation metrics that could be used are as follows:

1. Mean Absolute Error (MAE): MAE calculates the average absolute difference between the predicted and actual values. It is commonly employed for time series forecasting and regression tasks.
2. Root-Mean-Square Error (RMSE): RMSE computes the square root of the average squared difference between the predicted and actual values. It is similar to MAE but gives more weight to significant errors.
3. Precision and Recall: Precision and recall are valuable metrics for evaluating classification models. Precision measures the proportion of true positive predictions among all positive predictions, while recall gauges the proportion of true positive predictions among all actual positive cases.

### 5.2. Performance Analysis and Implementation

This section explains the experimental results obtained from the proposed RMSDC technique, as shown in Figure 6.

To evaluate the performance of our proposed model in comparison with recent frameworks for statistical downscaling of climate variables, we utilize two metrics: root-mean-square error (RMSE) and mean absolute difference (referred to as bias).

While our current research considers fifteen temporal inputs, incorporating additional temporal factors may enhance the model's performance. However, we are constrained in selecting a lag for this study due to computational limitations.

Figure 7 depicts the best forecasting results with the lowest RMSDC MSE. Despite the lowest MSE, Figure 7 shows a significant discrepancy at the beginning and halfway. However, as seen in Figure 7, the forecasting results are approaching the initial value as the period ends.

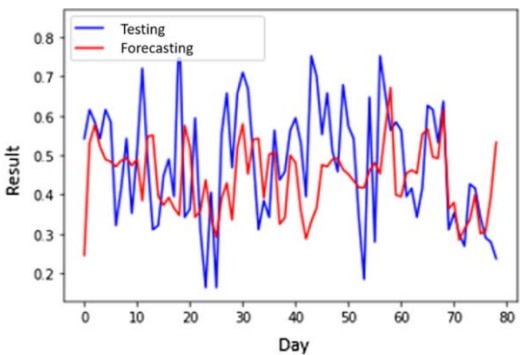

**Figure 7.** Testing and forecasting for Convolutional LSTM-based day.

We can contrast the plot behavior of training and testing losses. Figure 8 demonstrates that the training and testing losses decreased, with no rise in the testing loss at this level. Furthermore, because the testing loss was decreased, we could continue training without testing our training data. As a result, no overfitting happened.

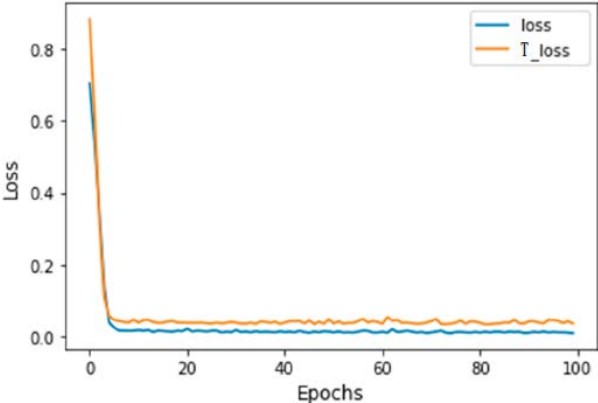

**Figure 8.** Training and loss validation for Convolutional LSTM-based model.

All of the climatic variables including friction, state, air temperature, latitude, longitude, water, speed, and the data from seven sensors are represented as channels on the left.

The number of kernels employed in a particular block is indicated by the numbers in each block's label, followed by the block's name and activation.

Baseline Models: The LSTM, RNN, CNN, CONV-LSTM, and RMSDC baseline models are used to compare the empirical results. All of the baseline models, including LSTM, which have produced better outcomes than in [56], had their hyperparameters optimized. The RNN, CNN, CONV-LSTM, and RMSDC models' optimum hidden state dimension values are 32, 32, and 64. This configuration has roughly 18,931; 19,190; and 58,774 trainable parameters for the LSTM, RNN, and CNN, respectively.

### 5.3. Results

We conducted tests to determine the optimal hyperparameter settings for training our RMSDC model. Our experiments found that using a hidden state dimension of 32 for both the encoder and decoder consistently produced superior results ($m = p$ for simplicity). We employed the Adam optimizer with a batch size 256 and a learning rate of 0.011. To

prevent overfitting, a dropout layer with a rate of 0.3 was added after each LSTM layer and each model was trained for 60 epochs.

For the observed friction value, water, air temperature, and sensors dataset, we performed trials to optimize the dimension reduction of the context vectors to q = 4 and set the input sequence length to Tx = 5. With these settings, the RMSDC model has approximately 24,833 trainable parameters. We utilized three metrics during the evaluation: mean absolute error (MAE), root-mean-square error (RMSE), and average. The models were trained using a NVIDIA Titan RTX GPU to ensure efficient processing.

The empirical results for the dataset consisting of pollutants, building and friction value, water, air temperature, and sensors are presented in Tables 2 and 3. These tables provide an overview of the performance of each model, including their respective training time per epoch, which indicates the time required to train the model once on the entire training set.

**Table 2.** Statistical error parameters of the proposed RMSDC technique for road maintenance based on the climate change in training and testing datasets at (Tx = 24, Ty = 4).

| Performance | | RMSE | MAE | Average | time |
|---|---|---|---|---|---|
| LSTM | Training | $4.04 \times 10^0$ | $5.43 \times 10^0$ | $7.78 \times 10^0$ | 6.2 |
| | Testing | $3.44 \times 10^0$ | $4.53 \times 10^0$ | $6.99 \times 10^0$ | 6.3 |
| RNN | Training | $0.9216 \times 10^0$ | $1.8921 \times 10^0$ | $3.81 \times 10^0$ | 2.5 |
| | Testing | $0.8124 \times 10^0$ | $1.223 \times 10^0$ | $3.01 \times 10^0$ | 2.3 |
| CNN | Training | $6.94 \times 10^0$ | $7.43 \times 10^0$ | $8.78 \times 10^0$ | 1.9 |
| | Testing | $5.83 \times 10^0$ | $6.95 \times 10^0$ | $9.88 \times 10^0$ | 1.8 |
| CONV-LSTM | Training | $1.52 \times 10^{-1}$ | $0.43 \times 10^0$ | $1.65 \times 10^0$ | 1.5 |
| | Testing | $1.31 \times 10^{-1}$ | $2.12 \times 10^{-1}$ | $5.27 \times 10^{-1}$ | 1.6 |
| RMSDC | Training | $0.0814 \times 10^{-1}$ | $0.1721 \times 10^{-1}$ | $0.9410 \times 10^{-1}$ | 0.98 |
| | Testing | $0.8813 \times 10^{-1}$ | $0.1913 \times 10^{-1}$ | $0.8812 \times 10^{-1}$ | 0.99 |

**Table 3.** Statistical error values of the proposed RMSDC technique for road examination based on the climate change in training and testing datasets at (Tx = 5, Ty = 2).

| Performance | | RMSE | MAE | Average | Time |
|---|---|---|---|---|---|
| LSTM | Training | 2.4444 | 2.5225 | $2.78 \times 10^0$ | 5.2 |
| | Testing | 1.5544 | 1.51228 | $5.99 \times 10^0$ | 5.5 |
| RNN | Training | 3.1776 | 4.0786 | $4.81 \times 10^0$ | 5.5 |
| | Testing | 3.1786 | 4.0795 | $4.01 \times 10^0$ | 3.7 |
| CNN | Training | 2.1611 | 2.0650 | $3.78 \times 10^0$ | 1.4 |
| | Testing | 2.1711 | 2.0660 | $2.88 \times 10^0$ | 1.5 |
| CONV-LSTM | Training | 2.0800 | 2.0478 | $2.841 \times 10^0$ | 1.12 |
| | Testing | 2.0900 | 2.0488 | $1.8812 \times 10^0$ | 1.2 |
| RMSDC | Training | 1.2534 | 2.0459 | $2.26 \times 10^0$ | 0.65 |
| | Testing | 2.0635 | 1.0559 | $2.17 \times 10^0$ | 0.72 |

Two essential indicators are used for performance evaluation: root-mean-square error (RMSE) and mean absolute difference. Our real-world analyses repeatedly show that RMSDC outperforms other techniques, with a 0.26 reduction in RMSE.

The quantitative findings highlight RMSDC's effectiveness in mitigating the effects of climate change on road networks. RMSDC improves traffic safety while simultaneously lowering costs and improving environmental sustainability by implementing proactive road repair measures. Furthermore, in developing countries with limited resources, climate adaptation of road networks is crucial, making RMSDC a sensible and practical solution for resilient road maintenance systems.

## 6. Conclusions

Road maintenance is a critical aspect of infrastructure management to ensure the safety and efficiency of transportation. However, predicting road maintenance needs accurately and efficiently remains a significant challenge, especially with the impact of climate change. In this study, we propose the RMSDC model with a ConvLSTM block to predict road maintenance needs using road maintenance systems and climate change data.

The proposed RMSDC model with ConvLSTM block uses historical data from the RCM411 sensors dataset to predict road maintenance needs based on spatial and temporal factors, including traffic volume and climate change data. The model uses an optimized algorithm for computing the root-mean-square deviation (RMSD) and a ConvLSTM block to capture the temporal dependencies and spatial correlations between the input features.

We demonstrate the effectiveness of the proposed RMSDC model in several experiments, including the prediction of road maintenance needs based on traffic volume and climate change data. The results show that the proposed RMSDC model with a ConvLSTM block outperforms existing methods in accuracy and efficiency.

The proposed RMSDC model with ConvLSTM block represents a significant advance in road maintenance and management, providing road maintenance organizations with a powerful tool to predict maintenance needs in the face of climate change. By leveraging the power of machine learning and deep learning, road maintenance organizations can improve the efficiency and effectiveness of their operations and provide better service to their communities.

## 7. Limitations and Future Works

The current work uses the RCM411 dataset; however, its size and diversity may be limited. A larger and more diversified dataset from various regions and climates would improve the RMSDC model's generalizability. Predictions of climate change are inherently uncertain. The accuracy of these projections and their capacity to accurately represent future climatic patterns may impact the RMSDC system's performance. The performance of the RMSDC model may be affected by the selection of hyperparameters. A more systematic examination of hyperparameter settings and optimization algorithms could be performed to discover the best configuration. The ConvLSTM architecture can be computationally demanding, mainly when dealing with massive datasets and sophisticated models. To reduce calculation time, efficient model architectures or hardware acceleration methods could be investigated.

Future research could concentrate on gathering more diverse and larger amounts of information from various regions and climates. Incorporating data from many sources and types of sensors would enrich the dataset even more. Future research can study the incorporation of additional external elements—such as traffic volume, road type, and building materials, which may affect road conditions—to improve the robustness of the RMSDC model. Ensemble learning techniques could be investigated to merge different RMSDC models or integrate other machine learning algorithms, thereby enhancing prediction accuracy and lowering uncertainty. Creating a real-time version of the RMSDC system would allow for continuous monitoring and adaptive road maintenance tactics, allowing quick reactions to changing weather conditions. Field trials of the RMSDC system on actual road networks can be conducted to examine its practical performance and effectiveness in real-world circumstances.

**Author Contributions:** Conceptualization H.E. and E.E.-S.; methodology, E.E.-S.; software, E.E.-S.; validation, A.A.; formal analysis, H.E. and E.E.-S.; writing—original draft preparation, E.E.-S. and H.E.; writing—review and editing, M.A. and M.I.I.; funding acquisition, M.A. All authors have read and agreed to the published version of the manuscript.

**Funding:** This work was supported by Researchers Supporting Project number (RSPD2023R636), King Saud University, Riyadh, Saudi Arabia.

**Institutional Review Board Statement:** Not applicable.

**Informed Consent Statement:** Not applicable.

**Data Availability Statement:** The data presented in this paper are available on request.

**Acknowledgments:** This work was supported by Researchers Supporting Project number (RSPD2023R636), King Saud University, Riyadh, Saudi Arabia.

**Conflicts of Interest:** The authors declare no conflict of interest.

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
