# Peer review of "A Deep Learning Technique to Improve Road Maintenance Systems Based on Climate Change"

_applsci, doi:10.3390/app13158899_

Round 1

Reviewer 1 Report

1. Please include the quantitative results in the Abstract.

2. Please explain the main differences between the proposed work and those published works. 

3. Most of the contents presented in Section 3 are the basic theory of CNN, RNN and LSTM. Most of these information are quite general and the contents should be reduced without affecting the conciseness. Furthermore, it is more appropriate to include these information under Literature Review.

4. The overall work flow of the RMSDC needs to be presented before the details in each stage are explained. 

5. Quality of flowchart in Fig. 4 needs to be improved further. Additionally, one block indicates "Excessive error!", which requires the attentions from the authors.

6. Authors need to look into the contents presented in Sections 4 and 5. Some of the information in Section 5 such as Section 5.1 and Section 5.3 are more relevant to the Section 4. Authors need to improve the organization of methodology flow. 

7. Please explain the limitations of current study and the future works that can be derived. 

Some grammatical errors and typos can be observed in the manuscript. Please proofread the manuscript again before resubmission.

Reviewer 2 Report

1.dataset is not described in detail.

2. Roads are varied in different types - highway, forest, hill, rural,etc. Each road has unique characteristics and patterns. But, it is not discussed.

3. Results are presented well. Spatial and temporal parameters can be discussed separately.

4. Algorithm for RMSDC Based on Multivariate can be added.

Proof reading is required

Reviewer 3 Report

The abstract mentions that the RMSDC technology is "capable of performing a variety of tasks at a level beyond human capability" and that "deep learning effectively creates intelligent road maintenance systems for climate adaptation in most countries." However, it lacks specific data or evidence to support these claims. Without empirical results or case studies, the claims seem speculative and less convincing.

Table 1 : we can make it more organized, provide clearer headings, and use proper formatting.

Section 3.1

Ambiguity in deep learning techniques: The paragraph states that deep learning techniques are used for classification, forecasting, and anomaly detection in time-series data. However, it does not specify which deep learning techniques are commonly employed for each task. Providing examples of specific algorithms or methods commonly used for time-series analysis would make the paragraph more informative.

For the rest of paper and the conclusion section

Enhanced clarity and readability by rephrasing certain sentences.

Added more specific information about the data sources and methods used.

Improved the flow of information to create a logical progression of ideas.

Expanded on the implications of the proposed model for road maintenance organizations and the communities they serve.

for example

"The RMSDC model with the ConvLSTM block represents a significant advancement in the field of road maintenance and management. This powerful tool equips road maintenance organizations with the ability to anticipate maintenance needs amid climate change challenges. By harnessing the potential of machine learning and deep learning, road maintenance organizations can enhance the efficiency and effectiveness of their operations, leading to improved services for their communities."

The paragraph provides an overview of some evaluation metrics that could be used for assessing the RMSDC model with ConvLSTM block in predicting road maintenance needs with climate change data. However, there are a few weaknesses that can be addressed: Lack of context: The paragraph mentions that the evaluation metrics depend on specific objectives, but it doesn't provide any information on what these objectives might be. Adding some context or examples of the specific objectives would make the paragraph more informative. Incompleteness: While the paragraph mentions three evaluation metrics (MAE, RMSE, Precision, and Recall), it could benefit from expanding on the suitability of these metrics for different aspects of the RMSDC model's performance evaluation. For example, what aspects of the model do these metrics assess, and in what situations would one be preferred over the other? Need for additional evaluation metrics: Evaluating the performance of a predictive model like RMSDC may require additional metrics to provide a comprehensive assessment. Metrics like F1 score, R-squared (for regression tasks), or Receiver Operating Characteristic (ROC) curve and Area Under the Curve (AUC) (for classification tasks) could be considered. Comparison with baseline: The paragraph doesn't mention whether these evaluation metrics will be compared against any baseline models or previous approaches. Including a comparison with baselines can help demonstrate the effectiveness and improvement of the proposed RMSDC model. A revised version of the paragraph considering these weaknesses could be: "The evaluation metrics for the RMSDC model with ConvLSTM block for predicting road maintenance needs with climate change data are contingent upon specific objectives and desired model performance. While several evaluation metrics can be considered, some suggested ones include: Mean Absolute Error (MAE): MAE calculates the average absolute difference between the predicted and actual values. This metric is commonly employed for time series forecasting and regression tasks, providing insights into the magnitude of errors. Root Mean Squared Error (RMSE): RMSE computes the square root of the average squared difference between the predicted and actual values. Similar to MAE, RMSE measures prediction accuracy, but it gives more weight to larger errors. This metric is valuable for understanding how the model performs with respect to outliers. Precision and Recall: Precision and recall are vital metrics for evaluating classification models, particularly if the RMSDC model is used for classifying road maintenance needs into different categories. Precision measures the proportion of true positive predictions among all positive predictions, while recall gauges the proportion of true positive predictions among all actual positive cases. These metrics are essential for assessing the model's performance in handling different classes effectively. It is important to note that these metrics provide valuable insights, but evaluating the RMSDC model's performance may also require considering additional metrics, such as the F1 score, R-squared (for regression tasks), or Receiver Operating Characteristic (ROC) curve and Area Under the Curve (AUC) (for classification tasks). Moreover, comparing the proposed RMSDC model's performance against baseline models or previous approaches would demonstrate its superiority and practical benefits."

Round 2

Reviewer 1 Report

Authors have addressed most comments given in previous review cycle. No further comments.

No major issues found on the quality of English langugage.

Reviewer 3 Report

The authors improved the paper and did all changes